# Study on Sulfide Oxidation in a Clay Matrix by the Hyphenated Method

**Eulalia Zumaquero \*** , **Jessica Gilabert, Eva María Díaz-Canales** , **María Fernanda Gazulla** and **María Pilar Gómez-Tena**

Instituto de Tecnología Cerámica, Asociación de Investigación de las Industrias Cerámicas, Universitat Jaume I, 12006 Castellón, Spain; jessica.gilabert@itc.uji.es (J.G.); evamaria.diaz@itc.uji.es (E.M.D.-C.); marife.gazulla@itc.uji.es (M.F.G.); pilar.gomez@itc.uji.es (M.P.G.-T.)

**\*** Correspondence: eulalia.zumaquero@itc.uji.es; Tel.: +34-964-342-424

**Abstract:** The requirement of a ceramic product with high technical and aesthetic performance makes it necessary to select and control raw materials to avoid losses caused by low-quality products. Many defects have their origin in impurities present in clay minerals such as sulfides and sulfates. It is important to study the oxidation, decomposition, and pyrolysis reactions that affect these minerals and their dependence on environment conditions (humidity and temperature) during the extraction and beneficiation of clay minerals in an open pit mine. The development of hyphenated techniques coupling mass spectrometry with a thermal analysis instrument provides information that is not available from either technique alone, such as decomposition behavior and the determination of emissions with a lower limit of detection. The evolution of sulfur dioxide from the oxidation of different sulfides provides information on the in situ oxidation and decomposition reactions that happen when a thermal treatment is applied. The results obtained show important differences in the reactions that take place in sulfides when they are stored under different environmental conditions. Specifically, the general tendency is that the sulfurs stored under high relative humidity show a decrease in the intensity of the emission as well as changes in the onset of the peak emission.

**Keywords:** evolved gas analysis; sulfur; pyrite; chalcopyrite

## 1. Introduction

Sulfides are a type of ore containing oxygen-free compounds of sulfur. They are found in sulfide-bearing rock in the ore body deep below the Earth's surface but their mineralogical structures, surface chemistry, and behavior depend on their biogenic formation and environmental conditions (magmatic ore deposits, hydrothermal deposits, sedimentary ore deposits, volcanic systems, etc.) [1].

Clay minerals have been mined from the Stone Ages and Middle Ages to present and they are among the most important minerals used by manufacturing industries. The environments of formation of these minerals include volcanic deposits, weathering rock formations, marine sediments, lake bottoms, and geothermal fields and most clay minerals form where rocks are in contact with water, air, or steam. Extensive alteration of rocks to clay minerals can produce relatively pure clay deposits that are of economic interest (for example, clays used in ceramics).

Typically, common clays and other raw ceramic materials are mined from open pits, and they are located near the processing plants to minimize production costs. Nevertheless, these open pit mines are exposed for long periods of time to very variable environmental conditions of humidity and temperature due to seasonal changes and extreme weather. Temperatures vary from 5 °C to 45 °C while the humidity ranges from 20% to 90% from the drier months to the wetter months, respectively. Additionally, some research suggests that many rock types containing kaolinite with different impurities such as sulfur compounds are sensitive to variations in the relative humidity and temperature [2]. Pyrite is a good

example of this because the pyrite oxidation process and the geochemical evolution of sulfur are controlled by reactivity reactions, which in turn are controlled by the surface structure and their dependence on humidity [3].

The composition and properties of clay raw materials are essential to controlling the industrial production process, since their mineralogy determines the firing and technical characteristics and properties of the end product, such as porosity, pyroclastic deformation, crack development, bloating, and black coring [4]. In fact, the presence of impurities in the raw materials is becoming more restrictive so a quality control procedure is essential to avoid the production of defects during the industrial ceramic process. These impurities can have their origin in carbonates, sulfides and sulfates, organic matter, and the presence of minerals with high iron content [5]. Many sulfide minerals are known to exist, and they are one of the most important groups of ore minerals because they are responsible for the concentration of a wide range of metals in mineable deposits. In fact, pyrite ($FeS_2$), pyrrhotite ($Fe_{1-x}S$), galena ($PbS$), sphalerite ($\beta$-$ZnS$), and chalcopyrite ($CuFeS_2$) are the most abundant sulfide minerals on Earth [1].

The major disadvantage of pyrites and other sulfides as impurities is that they produce specific defects in their crystallinity, which can produce changes in oxidation and decomposition processes. In general, defects produced in ceramic tiles due to sulfur compounds are normally associated with sulfates and sulfides and the presence of sulfates may form part of the mineralogy of the raw material or may come from the oxidation of sulfides. Although sulfates may be formed as minor products during the oxidation of pyrite, the presence of these compounds at low levels in a ceramic tile's composition can cause a defect in the final product because they decompose at temperatures lower than 900 °C.

The transformation process and the formation of sulfur compounds are affected by the reaction conditions, such as humidity, temperature, an oxidant or inert atmosphere, flow conditions, the particle size, and other structural parameters [6] that play a significant role in their interaction with the environment. A few studies have focused on the structural sensitivity of pyrite oxidation [7] and their results indicate that there is a surface structure–reactivity dependence of pyrite [8], which can be applied to understand the nature of pyrite oxidation and sulfur evolution. Along the same lines, there are numerous studies about the oxidation and decomposition processes [9] and the reaction mechanisms [10] of sulfides and pyrites not only in moist air [2] but in aqueous suspensions [11]. However, the influence of environmental conditions on the decomposition and oxidation processes in pyrite and chalcopyrite minerals when they are present in a clayey matrix in the form of impurities has not been studied.

Due to the heterogeneous nature of sulfur minerals, the complexity of the oxidation process, the presence of interference in a complex clayey matrix, and their very low concentration (usually less than 1%), it is not easy to use traditional techniques such as X-ray diffraction or wavelength-dispersive x-ray fluorescence. Indeed, some studies suggest that WD-XRF can only be used to measure the sulfur in samples having a chemical and mineralogical composition very close to that of the reference materials available for the calibration and/or validation of the methods used [12]. Moreover, when sulfur is present at a low concentration it is very difficult to discern whether the sulfur is present in the form of sulfides or sulfates [13], so the information WD-XRF provides is insufficient for the study of these impurities.

Thermal analysis methods, such as differential scanning calorimetry and thermogravimetry, are very powerful tools for investigating the thermal behavior of multi-mineral mixtures, geological samples [14], synthetic compounds [15], and natural materials [16]. Adding mass spectrometry for the analysis of the evolved gases during heating greatly increases the amount of information acquired. This makes evolved gas analysis (EGA) one of the most useful and versatile techniques available to determine the nature and number of volatile products formed during the thermal degradation of organic compounds such as coals [17] and inorganic materials such as pure minerals [18], clays [19], brick clays [20],

soils [21], and geological materials [22] and is extremely useful for the detection of small molecules [23] (for example, water, $CO_2$ [24], $SO_2$ [25], HCl, $NO_2$, and $NH_3$).

In this study, the evolution of sulfur species during thermal treatment was monitored and measured using TG-DSC and evolved gas analysis (EGA) with a quadrupole mass spectrometer. The objective was to analyze sulfur emissions during oxidation and decomposition processes when sulfides are found as impurities within a clayey matrix [26] with an emphasis on how they are affected by environmental conditions such as humidity and temperature and the particle size.

## 2. Materials and Methods

### 2.1. Characterization of Sulfides and the Clayey Matrix

The study was conducted with a commercial kaolin used in the manufacture of white-firing wall tiles and sulfur minerals that are considered to be impurities in ceramic raw materials. Their commercial references are as follows:

- Kaolin USP Ref: K1512 (Sigma-Aldrich);
- Pyrite, naturally occurring mineral, grains of approx. 1.5–4.8 mm, Ref: 42633 (Alfa-Aesar);
- Chalcopyrite, naturally occurring mineral, grains of approx. 1.5–4.8 mm, Ref: 42533 (Alfa Aesar).

The mineralogical composition of the samples was determined by X-ray powder diffraction (XRD, Bruker, Germany) with a Bruker D8Advance diffractometer with CuK$\alpha$ radiation, a Vantec-1 detector, and tube conditions of 30 kV and 40 mA. Under these conditions, the samples were measured between 5° and 90° (2θ) with a step size of 0.023° and a scan speed of 0.5 sec/step. The results of the mineralogical characterization with the identified crystalline structures of the materials are summarized in Table 1.

**Table 1.** Mineralogical and chemical composition of the samples.

| Sample | Crystalline Structure | Purity (%) | %S |
|---|---|---|---|
| Kaolin | Kaolinite ($Al_2Si_2O_5(OH)_4$) | <98 | 0.039 |
| Pyrite | Pyrite ($FeS_2$) | <98 | 52.3 |
| Chalcopyrite | Chalcopyrite ($CuFeS_2$) | <98 | 34.2 |

In order to determine the concentration of sulfur in the kaolin raw material, we used a wavelength dispersive X-ray fluorescence spectrometer (PANALYTICAL, model Axios WD-XRF, Almelo, The Netherlands), in which the sample was prepared in the form of pellets, the analytical line K$\alpha$ (2θ = 110.7°) with a $Ge_{111}$ crystal, a voltage of 30 kV, an intensity of 90 mA, and a flow detector. The sulfur concentration for this sample was 0.039% (by weight). Additionally, to guarantee the measurements' traceability, the following reference materials were used: STDS-1 Stream Sediment, NCS DC 60119 GBW 03118 Graphite Ore, and BCS-CRM 381 Basic Slag. It should be noted that the sulfur emission in kaolin was registered with a very low signal using the EGA technique because of the low percentage of sulfur in the material.

The sulfur emission analysis of different compositions was performed by EGA using a Netzsch STA 449 C Jupiter® simultaneous TG-DSC instrument equipped with a high-temperature tube furnace (SiC) and a Netzsch QMS 403 Aëolos® quadrupole mass spectrometer with an EI ion source and a mass range of up to 300 u. For the spectrometer coupling, a fused silica capillary with a small internal diameter connects the gas outlet of the thermal analyzer to the gas inlet of the QMS system. Both the adapter and the transfer line were heated to 300 °C to eliminate condensation. The measurements were performed in multiple ion detection mode (MID mode), which is very sensitive because only a few specific masses are detected. In fact, for $SO_2$ detection the fragments and molecular ions measured were m/z = 64, 48, and 32, which are the peaks in the mass spectrum with the greatest intensity. The optimized measurement conditions for sulfur dioxide determination by TGA-EGA were as follows [24]:

- an alumina crucible for sample sizes up to 0.2000 g;
- the sample mass ranged from 40 to 50 mg;
- a dynamic air atmosphere (20.9% $O_2$) with a flow rate of 50 mL·min$^{-1}$;
- helium as the protection gas at a flow rate of 25 mL min$^{-1}$;
- a maximum temperature of 1200 °C; and
- a heating rate of 10 °C min$^{-1}$.

The morphology and size of the particles of pyrite and chalcopyrite were studied with a FEG-SEM (QUANTA 200F, FEI Co, Hillsboro, USA). The powdered samples were deposited in a brass sample holder using a conducting carbon adhesive tape. Later, the prepared samples were observed and photographed with the backscattered electron signal of a field-emission gun environmental scanning electron microscope. The backscattered electron signal provides information on the topography and composition. The higher the average atomic number of the sample, the more intense is the signal, so that the lightest-colored areas contain the heaviest elements (a composition contrast).

The particle size distribution of different mineral fractions was determined using a MASTERSIZER 3000 laser diffraction instrument (MALVERN PANALYTICAL, Malvern, UK). The size distribution was measured using the parameters calculated using the MIE theory in order to interpret the light scattering signal collected by the detectors. The calculations were made considering a refractive index of 1.56 for pyrite and chalcopyrite and an absorption coefficient value of 0.1. The samples were mixed with water. After that, a 5-min ultrasonic bath was further applied in order to completely individualize the particles. Finally, the sample was mechanically stirred before being fed into the instrument.

### 2.2. Analysis Methodology

To conduct the study, the raw materials were dried at 110 °C for 2 h in a laboratory oven.

On the one hand, chalcopyrite was dry-milled and passed through a sieve with a nominal aperture of 45 microns. On the other hand, to study the effect of particle size, pyrite was sieved with a nominal aperture of 45 and 200 microns to obtain two different fractions. To determine the particle size and the morphology of each fraction, these samples were characterized with a FEG-SEM (Figure 1) and laser diffraction.

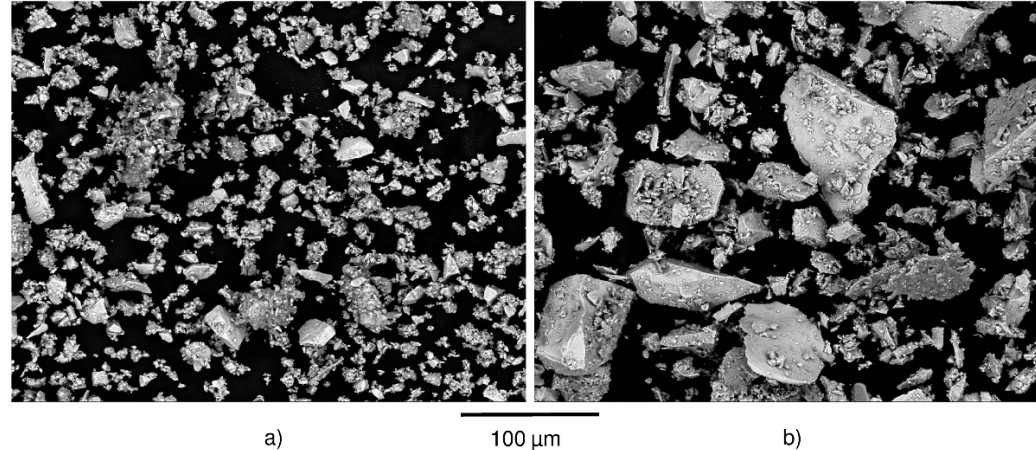

a)                    100 µm                    b)

**Figure 1.** SEM image (secondary electron signal ×800) of (**a**) the fine fraction of pyrite and (**b**) the coarse fraction of pyrite.

To prepare the compositions, sulfides were mixed in an agate mortar with kaolin in a percentage weight of 1% and homogenized in a HERZOG tungsten carbide ring mill.

To investigate the influence of temperature and humidity on the oxidation process, compositions were kept under different environmental conditions. The compositions were stored at 5 °C, 25 °C, and 45 °C and 20% and 90% relative humidity using different pieces

of equipment: a climatic chamber (model HC2020, HERAEUS); a laboratory fridge (model S75, NORCOOL); and a laboratory oven incubator (model INCUBATOR, RAYPA). Samples were maintained under these conditions for 1 month, 4 months, 6 months, and 1 year. These samples were periodically analyzed using the EGA technique to determine the types and amounts of weathering products that formed.

To perform the quantification of sulfur dioxide emissions with the EGA technique, the sulfur percentage was determined in the compositions and kaolin before and after the thermal treatment using WD-XRF and by preparing the samples in the form of pellets [27]. The results show that the calcined samples had percentages of sulfur that were below the detection limit (< 0.005% S); that is, during the heat treatment almost all of the sulfur present in the compositions had been emitted. Moreover, the area of the emission peaks of the curve obtained using the EGA technique was calculated. Thus, the relationship between emission area and ppm was obtained.

## 3. Results and Discussion

The mineralogy, particle size distribution, and influence of environmental conditions on oxidation processes were evaluated.

### 3.1. Properties of Sulfides

Sulfur compounds present in ceramic raw materials can decompose at around 500 °C (we would be talking about pyrite, chalcopyrite, or another sulfide) and at high temperatures (T > 800 °C), indicating the presence of sulfates [28].

Sulfur emissions from the mixture with 1% chalcopyrite (fraction < 45 μm, d50 = 6.2 μm) followed three different stages (Figure 2). The first sulfur dioxide emission started at 350 °C and had a maximum at 412 °C, the second emission had a maximum at 745 °C and seemed to be of minor importance, and the third emission started at 1150 °C.

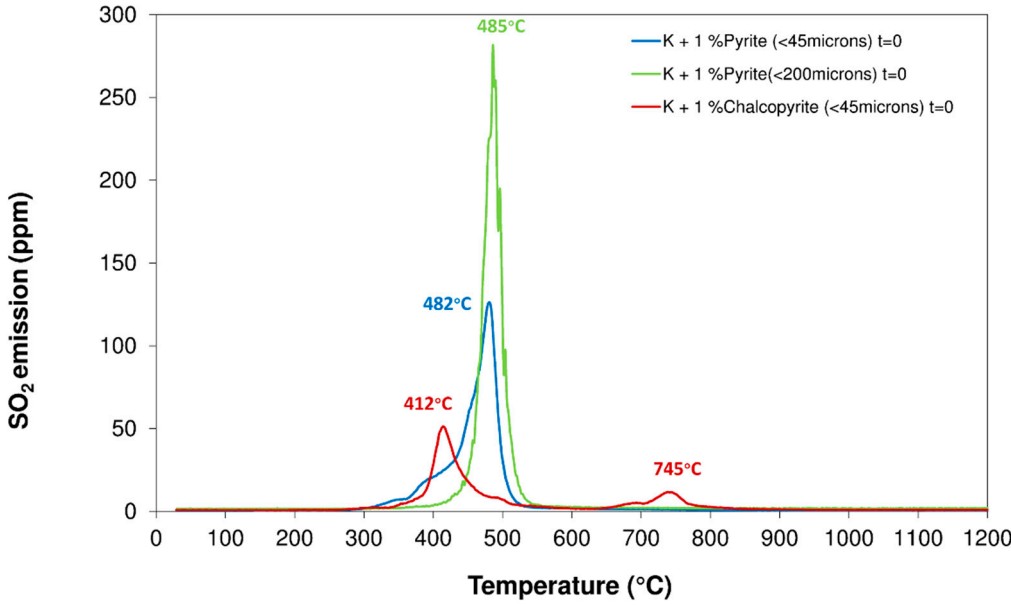

**Figure 2.** Evolution of the sulfur dioxide emissions of initial compositions.

The changes observed were mainly attributed to either the direct formation of copper sulfate and iron sulfate/oxide or the oxidation of the $Cu_2S/CuS$ and FeS obtained after the decomposition of chalcopyrite [29].

The composition with 1% pyrite content presented a first emission that was observed in the range of 300 °C to 900 °C (Figure 2). It was observed that the emission varied according to the particle size of the mineral added. The fine particle fraction of pyrite (fraction < 45 μm, d50 = 4.5 μm) presented an emission that started at around 320 °C

with a maximum at 482 °C and ended at around 540 °C. However, the coarser fraction of pyrite (fraction < 200 μm, d50 = 96.5 μm) began to emit sulfur at around 450 °C with a maximum at 485 °C and did not end this emission until it reached almost 550 °C. This emission corresponded to the oxidation and decomposition of sulfides. A second emission was observed from 1150 °C, indicating that the emission of sulfur had not finished when the heating cycle ended at 1200 °C, and this emission was due to sulfate decomposition.

### 3.2. Influence of Humidity and Temperature

3.2.1. Chalcopyrite

The formation of a sulfate in an oxidant atmosphere is an exothermic process, but the reaction is sluggish and takes place over a wide range of temperatures depending on the sulfide, the stoichiometry of the formed compound, and other factors [30] such as particle size (sulfate formation increases as particle size decreases), heating rate (sulfate formation increases with low heating rates), and morphology (increasing the mass size gas diffusion mechanism inhibits sulfate formation).

The evolved gas analysis results show that chalcopyrite had different oxidation and decomposition processes when subjected to different environmental conditions.

In the case of the fine fraction of chalcopyrite, the sulfur dioxide emission observed was a consequence of two important processes. The most important emission that took place in the temperature range of 300 °C and 600 °C was due to the oxidation of chalcopyrite to give $CuSO_4$ and $FeSO_4$. The second emission was registered in two steps: in the first step, the $FeSO_4$ that formed previously decomposed to give $Fe_2O_3$; and in the second step, some $CuSO_4$ decomposed to give $CuO$ with the emission of $SO_2$ in both cases [28].

$$4CuFeS_2 \text{ (s)} \rightarrow 4FeS \text{ (s)} + 2Cu_2S \text{ (s)} + S_2 \text{ (g)} \tag{1}$$

$$S_2 \text{ (g)} + 2O_2 \text{ (g)} \rightleftarrows 2SO_2 \text{ (g)} \tag{2}$$

$$FeS \text{ (s)} + 2O_2 \text{ (g)} \rightarrow FeSO_4 \text{ (s)} \tag{3}$$

$$Cu_2S \text{ (s)} + SO_2 \text{(g)} + 3O_2 \text{ (g)} \rightarrow 2CuSO_4 \text{ (s)} \tag{4}$$

$$CuSO_4 \text{ (s)} \rightarrow CuO \text{ (s)} + SO_3 \text{ (g)} \tag{5}$$

$$FeSO_4 \text{ (s)} \rightarrow FeO \text{ (s)} + SO_3 \text{ (g)} \tag{6}$$

$$4FeO\text{(s)} + O_2\text{(g)} \rightarrow 2Fe_2O_3 \text{ (g)} \tag{7}$$

As shown in Figure 3, the general trend in the emission curve was that the samples stored at high relative humidity showed a decrease in the intensity of the emission as well as an advance in the onset of the peak emission. The $SO_2$ emissions showed significant changes in their behavior in the oxidation of chalcopyrite and in the sulfate decomposition. By calculating the area of the emission peak in the range of 200 °C to 600 °C, 600 °C to 900 °C, and 900°C to 1200 °C, it was observed that the emission areas of samples stored at dry ambient temperature had not been modified. However, when the samples were stored at high relative humidity, there were important differences. The areas corresponding to the range 200 °C to 600 °C decreased over time, and it was still more pronounced for the samples stored at 45 °C with a reduction of 50% of the $SO_2$ emission. On the other hand, the calculated areas in the range from 600 °C to 900 °C and from 900 °C to 1200 °C increased over time and this effect was much more intense for samples stored at 45 °C (Figures 4–6). This fact could possibly be explained by the oxidation of the sulfide and the formation of sulfates when the surface particles were exposed to a humid air atmosphere.

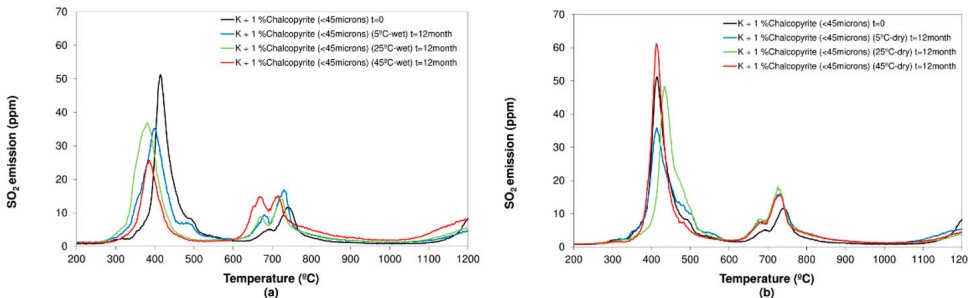

**Figure 3.** Evolution of the sulfur dioxide emission of the chalcopyrite composition in (**a**) a wet environment and (**b**) a dry environment.

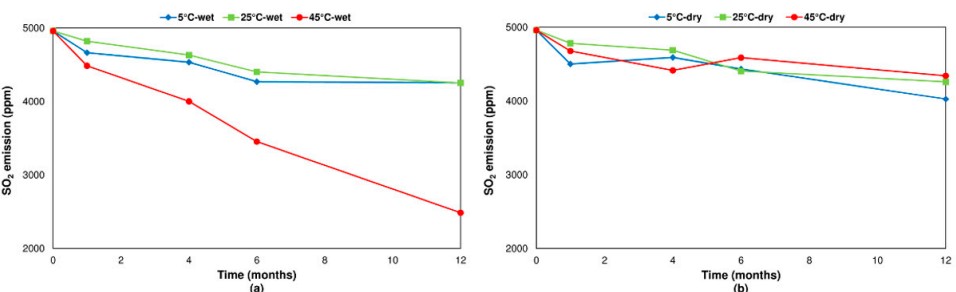

**Figure 4.** Sulfur dioxide emission area of the chalcopyrite composition at 200–600 °C in (**a**) a wet environment and (**b**) a dry environment.

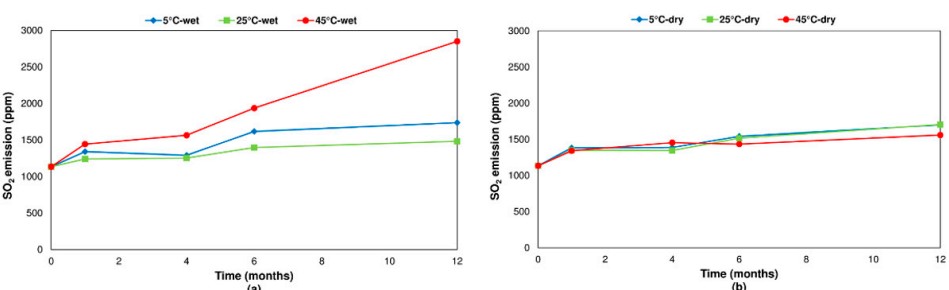

**Figure 5.** Sulfur dioxide emission area of the chalcopyrite composition at 600–900 °C in (**a**) a wet environment and (**b**) a dry environment.

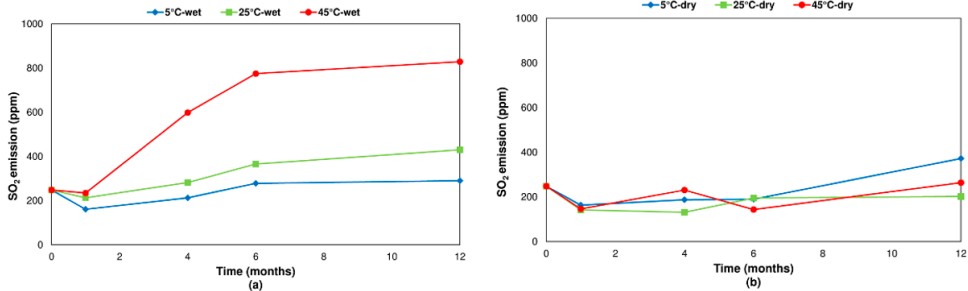

**Figure 6.** Sulfur dioxide emission area of the chalcopyrite composition at 900–1200 °C in a (**a**) wet environment and (**b**) a dry environment.

### 3.2.2. Pyrite

In the initial stages of the pyrite oxidation process, the oxygen, in the presence of water molecules, reacts on the surface, leading to the formation of sulfate and Fe (II) species. It is important to point out that many different mechanisms can contribute to the global reaction. The presence of different kinds of defects and impurities might have an important role in the mechanism.

Pyrite's major surface crystallographic planes are (100), (111), (110), and (210), but (100) is considered to be the most stable. Some authors [31] have studied the evolution mechanism of $SO_x$ formation during pyrite oxidation. The results showed that a chemisorption mechanism was responsible for $O_2$, $SO_2$, and $SO_3$ adsorption on the $FeS_2$ surface. In the same way, the oxidation of the bulk $FeS_2$ layer was controlled by a four-step process: bulk lattice S migration; lattice S oxidation; $SO_2$ desorption; and surface O atom deposition.

This might be because the oxidation and decomposition of the pyrite may produce changes in the structure of the surface of the pyrite crystal during the oxidation process that occurs on the surface of it. These reactions and their mechanisms are complex, and they are influenced by the presence of impurities, the purity of the mineral, and defects in the structure. Several studies on water adsorption on the pyrite surface have already been reported. In these studies, there is a consensus that water molecules adsorb molecularly over the Fe (II) sites on the surface. The research performed by Campos (2016) [10] shows that, in presence of water and oxygen, structural modifications occur on the surface of the pyrite, causing a rupture of the Fe–S bonds that facilitates the formation of hydroxides on the surface, and therefore a change in and relaxation of the structure affecting the oxidation process of pyrite.

Furthermore, Nesbitt et al. showed that the formation of sulfates occurs after a long period of surface exposure to a humid air atmosphere [32] and the layer formed due to the oxidation reaction can passivate against further oxidation. Different products are involved in the oxidation reaction [33], such as sulfate, iron oxy-hydroxide species, and possibly elemental sulfur and polysulfide [34]. The formation of "islands" of surface products provides further evidence for the concept of reactive and less-reactive surface areas [35].

Regarding this work, compositions were first prepared from 1% by weight of a fine fraction of pyrite. To study the effect of storage on the kinetics of sulfide oxidation [36], these compositions were stored for 12 months under different environmental conditions of humidity and temperature. These samples were periodically analyzed using the EGA technique. The results show the same behavior that was observed after one, four, six, and twelve months of aging. By calculating the area of the peak emission in the range of 200 °C to 600 °C, 600 °C to 900 °C, and 900 °C to 1200 °C, it was observed that the $SO_2$ emissions of samples stored at dry ambient temperature had not significantly changed (Figure 7). However, when the samples were stored at high humidity, we observed a reduction in the sulfur dioxide emission in the range of temperatures from 200 °C to 600 °C, where pyrite was oxidizing and decomposing. This effect occurred in the three tested temperature conditions (5 °C, 25 °C, and 45 °C), with a reduction in the $SO_2$ emission of 30% when the sample was stored at 45 °C. The opposite effect happened in the range 600 °C to 900 °C and 900 °C to 1200 °C, that is, an increase in the $SO_2$ emission was produced due to sulfate formation and decomposition (Figures 8 and 9).

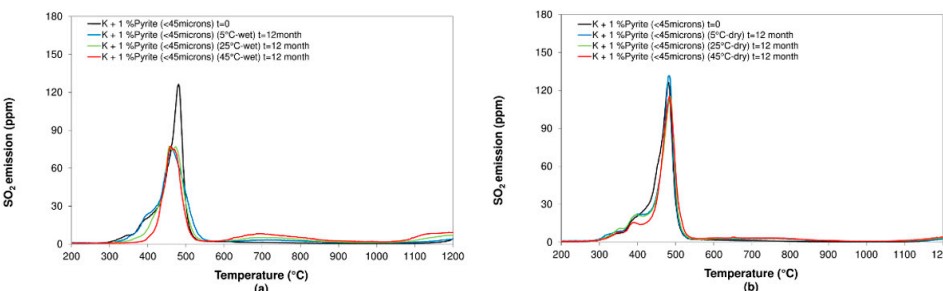

**Figure 7.** Evolution of the sulfur dioxide emission of the pyrite composition in (**a**) a wet environment and (**b**) a dry environment.

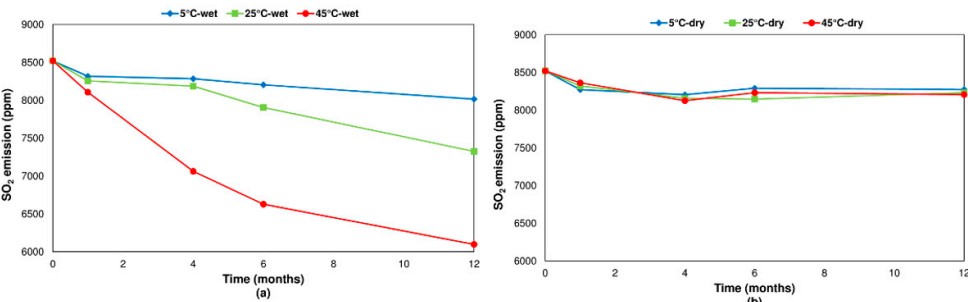

**Figure 8.** Sulfur dioxide emission area of pyrite (the fine fraction) at 200–600 °C in (**a**) a wet environment and (**b**) a dry environment.

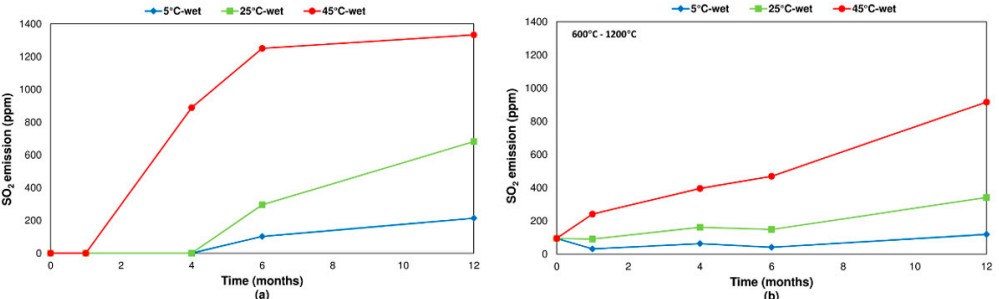

**Figure 9.** Sulfur dioxide emission area of pyrite (the fine fraction) (**a**) in the range 600–900 °C and a wet environment and (**b**) in the range 900–1200 °C and a wet environment.

### 3.3. Influence of Particle Size Distribution

Some authors affirm that the oxidation process of pyrite and other sulfides is influenced by the particle size distribution [37]. First, the small particles have a higher surface area than the large particles, so small pyrite particles have a larger pyrite–oxidant interfacial area available for oxidation and tend to react earlier than the large particles. Furthermore, less distance is required for oxygen diffusion into small particles than for oxygen diffusion into large particles. Consequently, the oxidation process is much faster in small particles than in large ones. Relating to sulfate formation, an increase in particle size could decrease the formation of the iron sulfate. On the other side, the sulfide's surface chemistry is particularly important because of its relevance to the oxidation and breakdown of sulfide minerals and to the processing of mined ores. In the case of pyrite, the oxidation process is related to the reactivity and surface chemistry.

The reaction mechanism of pyrite oxidation has been discussed extensively by many authors [38]. In an oxidant atmosphere, pyrite will be oxidized to form different compounds, such as magnetite ($Fe_3O_4$), hematite ($Fe_2O_3$), iron (ferric or ferrous) sulfate ($Fe_2(SO_4)_3$, $FeSO_4$) [39], and sulfur dioxide ($SO_2$). Considering only the main reactions and the related final products [9], the oxidation process can be followed by two different stages depending on whether the oxidation occurs directly (Equations (8) and (9)) or by the formation and decomposition of sulfates (Equations (10)–(14)). As has been indicated above, oxidation reactions can be described according to the following overall reactions when pyrite is heated:

$$2FeS_2 \text{ (s)} + 5.5O_2 \text{ (g)} \rightarrow Fe_2O_3 \text{ (s)} + 4SO_2 \text{ (g)} \tag{8}$$

$$SO_2 \text{ (g)} + 0.5O_2 \text{ (g)} \rightleftarrows SO_3 \text{ (g)} \tag{9}$$

$$2FeS_2 \text{ (s)} + 7O_2 \text{ (g)} \rightarrow Fe_2(SO_4)_3 \text{ (s)} + SO_2 \text{ (g)} \tag{10}$$

$$FeS_2 \text{ (s)} + 3O_2 \text{ (g)} \rightarrow FeSO_4 \text{ (s)} + SO_2 \text{ (g)} \tag{11}$$

$$2FeSO_4 \text{ (s)} \rightleftarrows Fe_2O_3 \text{ (s)} + SO_3 \text{ (g)} + SO_2 \text{ (g)} \tag{12}$$

$$Fe_2(SO_4)_3 \text{ (s)} \rightleftarrows Fe_2O_3 \text{ (s)} + 3SO_3 \text{ (g)} \tag{13}$$

$$SO_2 \text{ (g)} + 0.5O_2 \text{ (g)} \rightleftarrows SO_3 \text{ (g)} \tag{14}$$

To study the effect of particle size, the coarser fraction of pyrite was analyzed in the same way as the fine fraction. The results obtained in the EGA analysis of the compositions with the coarser fraction of pyrite (> 200 microns) show that the oxidation process of pyrite was very susceptible to the experimental conditions. Consequently, a change in the particle size distribution of the mineral could affect the evolution of the sulfur emission.

The experiments carried out in a humid environment showed a small reduction in the emission of $SO_2$ for samples stored at different temperatures (less than a 5% reduction). However, in samples stored in dry environments, no significant differences were observed (Figures 10 and 11).

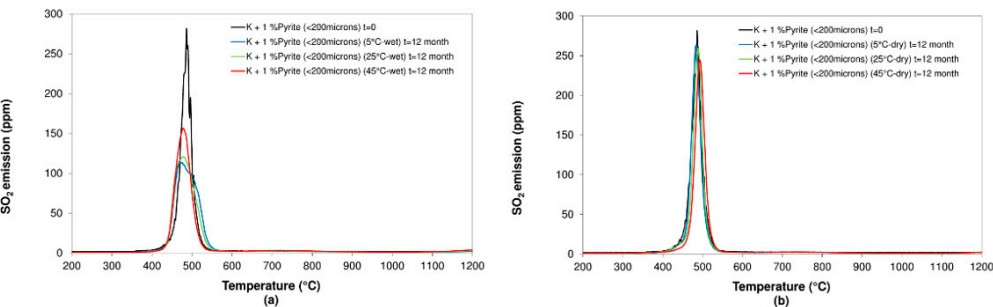

**Figure 10.** Evolution of sulfur dioxide emission of the coarser fraction of the pyrite composition in (**a**) a wet environment and (**b**) a dry environment.

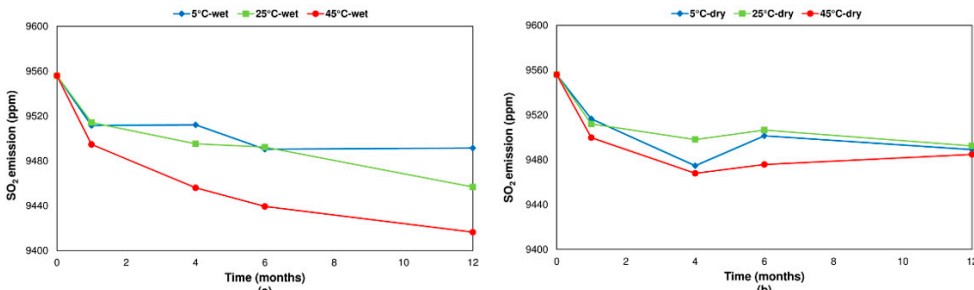

**Figure 11.** Sulfur dioxide emission area of pyrite (the coarser fraction) at 200–600 °C in (**a**) a wet environment and (**b**) a dry environment.

Comparing the results obtained with the coarse fraction and the fine fraction, a different behavior was observed. For example, comparing the experiments carried out under ambient temperature and 90% relative humidity conditions, it was noted that for the finer pyrite fraction the emission started at around 350 °C, while the coarse fraction began to emit sulfur dioxide at 400 °C. In addition, the composition prepared with the coarse fraction of pyrite showed a $SO_2$ emission with a narrower and more intense peak (Figure 12).

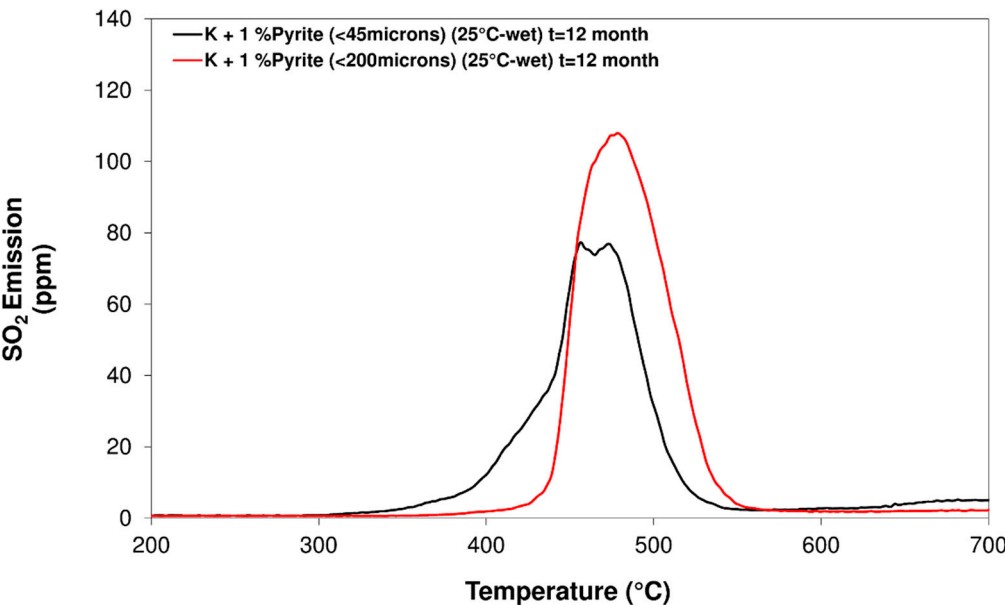

**Figure 12.** Comparison of the sulfur dioxide emissions of the coarse and fine fractions of pyrite (wet and dry environments at ambient temperature).

### 4. Conclusions

In this study, evolved gas analysis and thermal analysis with mass spectrometry were used to analyze differences in the behavior of the oxidation and decomposition of different sulfides and the influence of environmental conditions when they are present in a very low concentration in a clayey matrix. The results can be summarized as follows.

- The EGA technique allows us to register the sulfur dioxide emissions of minerals such as chalcopyrite and pyrite when applying a heat treatment. Different $SO_2$ emissions were observed due to the oxidation and decomposition stages of sulfides and sulfates formed during the pyrolysis process.
- By using hyphenated techniques, it is possible to detect, distinguish, and quantify sulfides in clayey matrices even when these are present in low concentrations.
- Concerning the influence of environmental conditions on the oxidation and decomposition processes, it was observed that samples stored at dry ambient temperature do not show significant differences. However, when the samples were stored in a high humidity environment, we observed a reduction in the sulfur dioxide emission in the range of temperatures from 200 °C to 600 °C, where sulfides are oxidizing and decomposing. On the other hand, an increase in $SO_2$ emissions was registered in the range of temperatures from 600 °C to 1200 °C because of sulfate decomposition. These effects are significantly more intense when samples are stored at a higher temperature (45 °C).
- Regarding the influence of the particle size, it was found, by observing changes in $SO_2$ emission profiles, that the reactivity and oxidation process of pyrite is related to the particle size. However, when changing the particle size of the pyrite, no notable differences were observed in the total $SO_2$ emission independently of the environmental conditions of temperature and humidity.
- To avoid defects derived from the industrial process, not only the concentration of sulfur but also the mineralogy and the environmental conditions to which the raw materials are exposed must be considered. This is because the environmental conditions of humidity and temperature have an important influence on the oxidation and degradation processes of the sulfides present in the raw ceramic materials.

**Author Contributions:** The authors conducted the following tasks in the research, in addition to successive revisions of the manuscript: conceptualization, E.Z. and M.P.G.-T.; funding acquisition, J.G. and M.P.G.-T.; investigation, E.Z. and M.P.G.-T.; methodology, E.Z., E.M.D.-C., and M.F.G.; project administration, J.G. and M.P.G.-T.; resources, E.Z.; supervision, E.Z. and M.P.G.-T.; validation, J.G., E.M.D.-C., and M.F.G.; writing—original draft, E.Z. and M.P.G.-T.; writing—review & editing, E.Z. and M.P.G.-T. All authors have read and agreed to the published version of the manuscript.

**Funding:** This study was co-funded by the Valencian Institute of Business Competitiveness (IVACE) and the FEDER Funds within the FEDER Operational Program of the Valencian Region 2014–2020, through projects IMDEEA/2019/28 and IMDEEA/2020/80.

**Conflicts of Interest:** The authors declare no conflict of interest. The funders had no role in the design of the study; in the collection, analyses, or interpretation of data; in the writing of the manuscript; or in the decision to publish the results.

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
