# Peer review of "Study on Sulfide Oxidation in a Clay Matrix by the Hyphenated Method"

_minerals, doi:10.3390/min11101121_

Round 1
Reviewer 1 Report
The paper called "Study of sulfide oxidation in clayey matrices using hyphenated techniques" is simple, but well structured, the methodology is easy to understand, and the results and conclusions are very clear and precise. I recommend the same for publication after minor revisions
Details to improve:
Throughout the document, justify the texts (looks messy)
Throughout the document change unit, for example: 5 to 45 ° C
Table 1. Can you provide more information about it? The% of each element in the compound? This information that is provided, I consider that it is not very relevant for the manuscript. I need this to be clarified or improved accordingly.
Author Response
Throughout the document, justify the texts (looks messy): The authors have changed this recommendation.
Throughout the document change unit, for example: 5 to 45 ° C: The authors have changed the format of units according to "Guide for the use of the International System of Units (SI)" published by NIST.
Table 1. Can you provide more information about it? The% of each element in the compound? This information that is provided, I consider that it is not very relevant for the manuscript. I need this to be clarified or improved accordingly. The authors have made more detailed information available in table 1.
Reviewer 2 Report
The paper deals with the problem of sulfide oxidation in clayey matrices. To this purpose Authors used hyphenated techniques. This joins thermal analysis with mass spectrometry. This allows to obtain full spectrum of the results, which are not possible to be achieved when only one of these methods is applied.
Paper consists of four parts. The first one presents the introduction. Here, the review of references is provided. In my opinion, a little bit too short and too general. Second chapter describes materials and methods. Authors used kaolin as clayey matrix as well pyrite and chalcopyrite as sulfides. Various measuring techniques were provided to obtain the results which were presented and discussed in chapter three. Discussion was done for chalcopyrite and pyrite cases concerning influence of humidity and temperature as well particle size distribution. Last chapter shows main conclusions.
Generally, the paper is ok, but introduction part should be rearranged by providing more references and their less general presentation. The main differences in the behavior of oxidation and decomposition of different sulfides were examined as well the influence of environmental conditions. There are errors in numeration, for example subchapter 3.1.1 should be numbered as 3.2.1 or subchapter 3.1.2 as 3.2.2. The presented results can be valuable in determination of ceramic products requirements.
Author Response
Introduction part should be rearranged by providing more references and their less general presentation. The authors consider that 23 bibliographical references related to the main objective of the research work are adequated for the introductory section. Further references are included in the methodology and results sections.
There are errors in numeration, for example subchapter 3.1.1 should be numbered as 3.2.1 or subchapter 3.1.2 as 3.2.2. The authors have corrected the errors in subchapter numeration.
Thank you for your comments.
Reviewer 3 Report
Manuscript ID: minerals-1409851
Title: Study of sulfide oxidation in clayey matrices using hyphenated techniques
Authors: Eulalia Zumaquero et al.
The article is well-formed research. In Section 2 the description of the analysis methods was written very detail. The results obtained are substantiated and performed at a good scientific level. There are small notes on the presentation of the results that can be quickly corrected. This research may be of interest to ceramics manufacturers.
The title must be improved. Study of sulfide oxidation in clay matrix by the hyphenated method.
Figure 1. The image clarity is very poor, although the authors characterize very large particles. Authors should improve the clarity of the SEM images.
Figure 2. Authors must use different colors for each curve.
Line 208. The authors write about the last stage of sulfur emission at 1150 ºC. however, Figure 1 shows only T = 200-900 ºC. Please, add data from 900 ºC to 1200 ºC.
Figures 2-9. Authors must use different colors for each curve.
Authors must add oxidation reactions for Chalcopyrite as like reactions for pyrite (line 334-335).
The article "Study of sulfide oxidation in clayey matrices using hyphenated techniques" corresponds to the level of Minerals and can be accepted after minor revision.
Author Response
The title must be improved. Study of sulfide oxidation in clay matrix by the hyphenated method. The authors have changed the title of the paper according to the reviewer's comment.
Figure 1. The image clarity is very poor, although the authors characterize very large particles. Authors should improve the clarity of the SEM images. The authors have modified the quality and resolution of the images provided by SEM.
Figure 2. Authors must use different colors for each curve. The authors have changed all figures containing curves using different colours to improve the interpretation of the results.
Line 208. The authors write about the last stage of sulfur emission at 1150 ºC. however, Figure 1 shows only T = 200-900 ºC. Please, add data from 900 ºC to 1200 ºC. The authors have extended the data up to 1200ºC, although in the temperature range 900ºC to 1200ºC no significant changes are observed.
Figures 2-9. Authors must use different colors for each curve. The authors have changed all figures containing curves using different colours to improve the interpretation of the results.
Authors must add oxidation reactions for Chalcopyrite as like reactions for pyrite (line 334-335). The authors have included the oxidation reactions of chalcopyrite.
Thank you for your comments.
Reviewer 4 Report
The manuscript is a study of sulfide oxidation in clayey matrices using hyphenated techniques. The scope of this article is consistent with the requirements of the Minerals Journal, but it requires minor revision in accordance with the comments below:
- Avoid lumping references as in: [6-8], [9-11], [12-14], [17-23]. Instead summarise the main contribution of each referenced paper in a separate sentence.
- The quality of figures 3-11 is poor. The figures are to small.
Author Response
Avoid lumping references as in: [6-8], [9-11], [12-14], [17-23]. Instead summarise the main contribution of each referenced paper in a separate sentence. The authors have modified the references, avoiding grouping references.
The quality of figures 3-11 is poor. The figures are to small.
Due to the large number of figures throughout the paper, the authors selected a small format in figures to improve comparison and interpretation of results in different environmental conditions.
The resolution of the figures has been increased and curves have been coloured. However, if the reviewer deems it appropriate, the size of the figures could be increased.